

# Interaction of a dinoflagellate neurotoxin with voltage-activated ion channels in a marine diatom

Sheila A. Kitchen[1], Andrea J. Bourdelais[2] and Alison R. Taylor[1]

[1] Department of Biology and Marine Biology, University of North Carolina Wilmington, Wilmington, NC, United States of America
[2] Center for Marine Science, University of North Carolina Wilmington, Wilmington, NC, United States of America

## ABSTRACT

**Background**. The potent neurotoxins produced by the harmful algal bloom species *Karenia brevis* are activators of sodium voltage-gated channels (VGC) in animals, resulting in altered channel kinetics and membrane hyperexcitability. Recent biophysical and genomic evidence supports widespread presence of homologous sodium ($Na^+$) and calcium ($Ca^{2+}$) permeable VGCs in unicellular algae, including marine phytoplankton. We therefore hypothesized that VGCs of these phytoplankton may be an allelopathic target for waterborne neurotoxins produced by *K. brevis* blooms that could lead to ion channel dysfunction and disruption of signaling in a similar manner to animal $Na^+$ VGCs.

**Methods**. We examined the interaction of brevetoxin-3 (PbTx-3), a *K. brevis* neurotoxin, with the $Na^+/Ca^{2+}$ VGC of the non-toxic diatom *Odontella sinensi*s using electrophysiology. Single electrode current- and voltage- clamp recordings from *O. sinensis* in the presence of PbTx-3 were used to examine the toxin's effect on voltage gated $Na^+/Ca^{2+}$ currents. *In silico* analysis was used to identify the putative PbTx binding site in the diatoms. We identified $Na^+/Ca^{2+}$ VCG homologs from the transcriptomes and genomes of 12 diatoms, including three transcripts from *O. sinensis* and aligned them with site-5 of $Na^+$ VGCs, previously identified as the PbTx binding site in animals.

**Results**. Up to 1 μM PbTx had no effect on diatom resting membrane potential or membrane excitability. The kinetics of fast inward $Na^+/Ca^{2+}$ currents that underlie diatom action potentials were also unaffected. However, the peak inward current was inhibited by 33%, delayed outward current was inhibited by 25%, and reversal potential of the currents shifted positive, indicating a change in permeability of the underlying channels. Sequence analysis showed a lack of conservation of the PbTx binding site in diatom VGC homologs, many of which share molecular features more similar to single-domain bacterial $Na^+/Ca^{2+}$ VGCs than the 4-domain eukaryote channels.

**Discussion**. Although membrane excitability and the kinetics of action potential currents were unaffected, the permeation of the channels underlying the diatom action potential was significantly altered in the presence of PbTx-3. However, at environmentally relevant concentrations the effects of PbTx- on diatom voltage activated currents and interference of cell signaling through this pathway may be limited. The relative insensitivity of phytoplankton VGCs may be due to divergence of site-5 (the putative PbTx binding site), and in some cases, such as *O. sinensis*, resistance

Corresponding author
Alison R. Taylor, taylora@uncw.edu

to toxin effects may be because of evolutionary loss of the 4-domain eukaryote channel, while retaining a single domain bacterial-like VGC that can substitute in the generation of fast action potentials.

## INTRODUCTION

Periodic algal blooms of the toxic dinoflagellate, *Karenia brevis* (Davis, 1948; Hansen & Moestrup, 2000), have been reported since the mid-1600s in the Gulf of Mexico, predominately along the west coast of Florida (*Steidinger, 2009*). These blooms result in massive marine animal mortalities (*Hackett et al., 2004*; *Pierce & Henry, 2008*; *Landsberg, Flewelling & Naar, 2009*) and also present problems for humans, where toxin exposure is a major public health concern because exposure though consumption of contaminated shellfish or inhalation of aerosolized toxins can cause adverse neurological and respiratory symptoms associated with neurotoxic shellfish poisoning (*Smayda, 1997*; *Van Dolah, 2000*; *Fleming et al., 2011*). The associated economic losses (*Hoagland et al., 2002*) and increased frequency and duration of harmful algal blooms (HABs) (*Paerl & Whitall, 1999*; *Van Dolah, 2000*) has led to increased efforts to understand the factors that regulate algal toxin production, bloom dynamics and toxin susceptibility (*Van Dolah, 2000*; *Pierce & Henry, 2008*; *Henrichs, Hetland & Campbell, 2015*; *Weisberg et al., 2016*). In spite of these efforts, the functional role for PbTx production by *K. brevis* remains poorly understood.

The potent, lipophilic neurotoxins produced by *K. brevis* are classified by their polyether ladder backbone chemical structures, PbTx-A (Pbtx-1, -7, -10) or PbTx-B (PbTx-2, -3, -6, -9). PbTx-1 and PbTx-2 are the intracellular parent compounds and most toxic, while the less toxic extracellular derivatives are produced through intermediary metabolism and environmental turnover (*Pierce & Henry, 2008*). The biosynthetic pathway for brevetoxin production is yet to be resolved, although cytosolic (*Van Dolah et al., 2013*), plastidic (*López-Legentil et al., 2010*; *Monroe et al., 2010*; *Van Dolah et al., 2013*), modular and single domain polyketide synthases (*Van Dolah et al., 2017*) may be involved in production of these secondary metabolites. Intracellular concentrations of PbTx can vary widely with bloom age, salinity, nutrient, light and clone type making bloom toxicity difficult to predict (*Brown et al., 2006*; *Lekan & Tomas, 2010*; *Corcoran, Richardson & Flewelling, 2014*). However, it only takes pM to nM concentrations of these potent toxins to be lethal in some vertebrate models (*Baden, 1989*; *Pierce & Henry, 2008*). Nevertheless, it is unlikely that the potent effect of PbTx on metazoans represents the primary functional role for toxin production.

Several studies have addressed whether these algal secondary metabolites underpin a chemically mediated allelopathy against grazers. Ingestion of toxic *K. brevis* cells by pelagic copepods (*Breier & Buskey, 2007*; *Cohen, Tester & Forward Jr, 2007*; *Hong et al., 2012*; *Lauritano et al., 2013*), rotifers (*Kubanek, Snell & Pirkle, 2007*), benthic amphipods
and urchins (*Sotka et al., 2009*) have variable effects on zooplankton growth or fecundity, with little evidence of an anti-predation mechanism. Studies have also investigated whether PbTxs confer an allelopathic advantage against other ecologically relevant targets such as co-occurring phytoplankton. Culture experiments have provided evidence for a wide range of interactions, from resistance to sublethality of phytoplankton grown with *K. brevis* or exposed to their purified toxins (*Freeberg, Marshall & Heyl, 1978*; *Kubanek et al., 2005*; *Myers et al., 2008*; *Prince, Myers & Kubanek, 2008*; *Prince et al., 2008*; *Poulson et al., 2010*; *Poulson-Ellestad et al., 2014a*; *Poulson-Ellestad et al., 2014b*). For example, *Freeberg, Marshall & Heyl (1978)* reported species-specific effects of *K. brevis* culture filtrates or cell extracts on cell growth of 18 different phytoplankton species encompassing four different phyla. As a follow up to this study, *Kubanek et al. (2005)* also observed species-specific effects on phytoplankton competitors when exposed to live *K. brevis* or their filtrates, bulk extracts or purified toxins. Most notable was the reduced growth of the diatoms *Amphora* sp., *Asterionellopis glacialis*, *Rhizosolenia cf. setigera*, *Skeletonema costatum* and *Thalassiosira* sp., and dinoflagellates *Akashiwo sanquinea*, *Peridinium* sp., and *Prorocentrum minimum* in the presence of live *K. brevis*, but not necessarily with purified PbTxs. On the other hand, other competitors such as the diatom *Odontella aurita* and chlorophyte *Chlorella capsulata* were unaffected by live *K. brevis*; however, their response to purified PbTx was not tested (*Kubanek et al., 2005*). Additional experiments have reported that extracts from natural blooms contain a mixture of other allelopathic compounds that reduce growth of some phytoplankton competitors more than the derivative forms of the extracellular PbTxs (*Prince, Myers & Kubanek, 2008*; *Poulson et al., 2010*; *Poulson-Ellestad et al., 2014b*). Although many interactions have been reported with PbTx derivatives or other allelopathic compounds, the underlying cellular mechanisms remain unknown.

In the case of animals, binding assays with PbTx show the primary molecular target is $Na^+$ VGCs (*Poli, Mende & Baden, 1986*; *Gawley et al., 1992*) critical for propagation of electrical signals in the neuromuscular system (*Baden, 1989*; *Gawley et al., 1992*; *Trainer, Baden & Catterall, 1994*; *Baden, Rein & Gawley, 1998*; *Cestèle & Catterall, 2000*; *Baden et al., 2005*). These metazoan $Na^+$ channels belong to a large eukaryote family of 4-domain VGCs (D1–D4), each composed of six transmembrane spanning segments (6TM) designated S1–S6 (*Cestèle & Catterall, 2000*; *Anderson, Roberts-Misterly & Greenberg, 2005*). The S4 segment of each domain contains positively charged arginine residues that act as voltage sensors for channel activation (*Yang & Horn, 1995*). Voltage dependent inactivation is controlled by a short intracellular loop connecting D3 and D4 that blocks the pore during depolarization (*Armstrong & Bezanilla, 1977*; *West et al., 1992*). The binding site of PbTx was first characterized in $Na^+$ VGC, type II ($Na_v1.2$) that are broadly distributed in neurons and highly sensitive to another neurotoxin, tetrodotoxin (*Trainer, Baden & Catterall, 1994*; *Lee & Ruben, 2008*). The PbTx receptor site-5 has two components, one region on the intracellular α-subunit (D1, S6) and one on the extracellular α-subunit (D4, S5) (*Trainer, Baden & Catterall, 1994*). The molecular model proposed (*Trainer, Baden & Catterall, 1994*) suggests that the hydrophobic PbTx interacts with both sites simultaneously by binding to the transmembrane interface and traversing the pore region of the $Na^+$ channel. Specific PbTx interaction with metazoan $Na^+$ VGCs results in: 1. a negative shift of channel

activation (*Huang, Wu & Baden, 1984*), 2. inhibition of channel inactivation (*Huang, Wu & Baden, 1984*; *Poli, Mende & Baden, 1986*; *Ulbricht, 2005*), and 3. an increase in Na$^+$ influx associated with prolonged membrane depolarization (*Atchison et al., 1986*; *Sheridan & Adler, 1989*). However, binding affinity and alteration of channel kinetics by PbTx varies depending on the derivative form of the toxin, tissue type exposed and species specificity (*Bottein Dechraoui & Ramsdell, 2003*; *Bottein Dechraoui, Wacksman & Ramsdell, 2006*).

Genomic and biophysical studies demonstrate that rapid membrane excitability associated with the Na$^+$/Ca$^{2+}$ VGCs, evolved prior to the divergence of Metazoa with representative 4-domain VGCs found in several unicellular protists including phytoplankton (*Armbrust et al., 2004*; *Verret et al., 2010*; *Liebeskind, Hillis & Zakon, 2011*; *Moran et al., 2015*). Evidence for the presence of such VGCs in several phytoplankton groups includes the dinoflagellates (*Eckert & Sibaoka, 1968*; *Ryan, Pepper & Campbell, 2014*), coccolithophores (*Taylor & Brownlee, 2003*), and diatoms (*Gradmann & Boyd, 1995*; *Taylor, 2009*). The marine diatom *Odontella sinensis* (Greville 1866; Grunow 1884) exhibits spontaneous and evoked action potentials with similar kinetic and pharmacological characteristics closely resembling Na$^+$/Ca$^{2+}$ VGC mediated action potentials of animals (*Taylor, 2009*). We therefore hypothesized that PbTx interacts with VGCs underlying phytoplankton membrane excitability in such a way as to disrupt sensory or signaling processes, thereby conferring a competitive advantage for the toxin producer. In order to test this hypothesis, we examined the effects of PbTx-3, the most abundant extracellular form of PbTx (*Lekan & Tomas, 2010*), on the voltage-activated Na$^+$/Ca$^{2+}$ current previously characterized in the diatom *O. sinensis* (*Taylor, 2009*).

## MATERIALS AND METHODS

### Phytoplankton cultures

*O. sinensis* (PLY606), obtained from Plymouth Culture Collection at the Marine Biological Association, UK, was batch cultured in 50 ml polystyrene vented flask (Cellstar, Fisher) with autoclaved, 0.2 μm filtered seawater enriched with f/2 nutrients, Guillard's vitamins and 2 mM NaHCO$_3$ (*Guillard & Ryther, 1962*). Cultures were maintained at 15 °C under 12:12 day:night cycle with 100 μmol m$^{-2}$ s$^{-1}$ light. Cells in mid-exponential phase were typically 50–100 μm long and 30–50 μm wide. All experiments were conducted using cells collected during exponential growth phase of the batch culture at a cell density of ∼10$^5$ ml$^{-1}$ determined by cell counts with a haemocytometer.

### Single electrode current and voltage clamp recordings and analysis

Single electrode current and voltage clamp experiments were conducted using methods previously described for *O. sinensis* (*Taylor, 2009*). Microelectrodes were fabricated from GC150F-10 thick walled borosilicate glass capillaries (Harvard Apparatus, Kent, UK) using a Flaming-Brown Micropipette Puller P-97 (Sutter Instruments, Petaluma, CA, USA) and coated with beeswax to reduce stray capacitance. Electrodes were filled with 1 M KCl and those with tip resistance between 8–14 M Ω were used. Electrodes were inserted into the headstage of an Axoclamp 900A amplifier (Axon Instruments, Union City, CA, USA) mounted on a Sutter MP285 motorized micromanipulator (Sutter Instruments, Petaluma,

CA, USA) that was attached to an inverted microscope (Nikon Diaphot, Nikon, Melville, NY, USA)

The single electrode current clamp and voltage clamp experiments were controlled and acquired with a PC connected to the amplifier via a Digidata 1200A interface (Axon Instruments, Union City, CA, USA). An external gain was applied prior to A–D conversion using a DC amplifier (LBF-100B; Warner Instrument Corps, Hamden, CT, USA). Data were acquired using Clampex v10.2 and all analysis was conducted offline with Clampfit v10.2 software (Axon Instruments, Union City, CA, USA).

We observed that *O. sinensis* takes up fluorescently labeled PbTx in membranes and lipid bodies (S Kitchen, 2015, unpublished data), indicating that these lipophilic molecules interact with diatom cell membranes and can accumulate. A 1 μM concentration of PbTx-3 was used in all electrophysiological experiments because 400 nM labeled-PbTx-B showed significant staining in 30 min. The higher concentration of unreacted PbTx-3 therefore ensured any effect should be detected during a typical recording of about 30–45 min and falls within the range (500 nM–2.8 μM) of the original experiments that described the interaction of PbTx with $Na^+$ VGCs (*Huang, Wu & Baden, 1984*; *Atchison et al., 1986*; *Sheridan & Adler, 1989*). Purified PbTx-3 provided by Dr. D.G. Baden of the University of North Carolina Wilmington Center for Marine Science was prepared as a 1 mM stock solution in DMSO and stored at −20 °C. A working solution was made up in ASW to a concentration of 1 μM immediately prior to experiments. Lidocaine (Sigma, St. Louis, MO, USA), a known inhibitor of *O. sinensis* action potentials (*Taylor, 2009*), was prepared as a 1 M stock in ASW with a working solution of 1 mM in ASW and stored at 4 °C.

*O. sinensis* cells were plated onto poly-L-lysine coated 35 mm glass bottom dishes (MatTek Corporation, Ashland, OR, USA) with a final bath volume of 1 ml and perfused with ASW from a gravity fed reservoir (1.5 ml min⁻¹). For both the current and voltage clamp experiments, initial control recordings were conducted under ASW perfusion. The perfusion was then stopped and a bolus addition of ASW ($n = 7$), 1 μM PbTx-3 ($n = 14$) or 1 mM lidocaine ($n = 4$) was added to a final bath volume of 1 ml and the cells incubated for at least 15 min before acquiring recordings in the presence of the test compound. Post-treatment washes by perfusion with fresh ASW were applied for an additional 15–30 min. All experiments were conducted at room temperature (20–22 °C).

## Bioinformatic analysis of the PbTx-3 binding site

At the time of this study, two diatom genome assemblies were available, one of the polar centric diatom *Thalassiosira pseudonana* (class Mediophyceae) and raphid pennate diatom *Phaeodactylum tricornutum* (class Bacillariophyceae, (*Armbrust et al., 2004*; *Bowler et al., 2008*)). In addition ten diatom transcriptomes (see Table S1) were available either through the Marine Microbial Eukaryote Transcriptome Sequencing Project (*Keeling et al., 2014*) deposited at iMicrobe (http://imicrobe.us, accessed June 14, 2015) or National Center for Biotechnology Information (NCBI), of which five belong to the class Bacillariophyceae, one belongs to class Coscinodiscophyceae (although the current phylogenetic placement of *Attheya* sp. is unresolved (*Parks, Wickett & Alverson, 2017*)) and four belong to the class Mediophyceae, including *O. sinensis* and its sister species *O. aurita*.

To determine the conservation of the predicted PbTx binding sites (*Trainer, Baden & Catterall, 1994*) in diatoms and other photosynthetic protists, we searched available sequence resources to find homologous single-domain or 4-domain VGCs in diatoms and other representative taxa. BLAST searches (BLASTp or tBLASTn version 2.2.31+) (*Altschul et al., 1990*) were performed with 4-domain VGCs protein sequences from a squid (*Heterololigo bleekeri*; NCBI: BAA03398.1), a rat (*Rattus norvegicus* $Na_v$ 1.4; NCBI: AAA41682.1) and the diatom *T. pseudonana* (NCBI: XP_002289136.1; previously reported in *Verret et al. (2010)* and *Moran et al. (2015)*), and single-domain VGC protein from the bacterium *Bacillus halodurans* C-125 (NCBI: BAB05220.1) against NCBI or local BLAST databases (Table S1) built from the diatom transcriptomes. Significant thresholds for sequence similarity were set to an $E$-value $<1e^{-10}$ in the large database (NCBI) or a bit-score $\geq 50$ in the small databases (local). Multiple sequence alignments were constructed from the top scoring sequences using MUSCLE (*Edgar, 2004*) in the software package Geneious v8.0.3 (*Kearse et al., 2012*). The voltage sensor, selectivity-filter and inactivation gate are well-characterized structural motifs of $Na^+$ VGCs (*West et al., 1992*; *Tsushima, Li & Backx, 1997*; *Catterall, 2000*) that were used as features to manually curate the orthologs from the sequence alignments. The final set of sequences with hallmark features of a $Na^+$ VGC are listed in Table S1. The PbTx-3 binding site originally identified in $Na_v1.2$ (NCBI: CAA27287.1) was used for sequence comparison to the $Na^+$ VGCs found in this study (Table S1). The NaChBac channel is predicted to form a tetramer of four identical domains (*Ren et al., 2001*). To compare the PbTx binding site to single-domain VGCs found in *O. sinensis*, we concatenated the single domain of the protein sequence #4108 into a hypothetical homotetramer. Visualization of protein structure between bacterial NaChBac (NCBI: BAB05220.1), diatoms *O. sinensis* (iMicrobe: #4108), *P. tricornutum* (NCBI: XP_002186055.1) and *T. pseudonana* (NCBI: XP_002289136.1 and XP_002287819.1), dinoflagellate *K. brevis* (iMicrobe: SPI 287987) and $Na_v1.2$ (NCBI: CAA27287.1) was made using Domain Graph (DOG) v2.0 (*Ren et al., 2009*).

## Statistical analysis

Quantitative electrophysiological parameters are presented as the mean $\pm$ SE. Data for control and PbTx-3 treatments were analyzed using a paired Student's $T$-test with an alpha level of $\leq 0.05$ chosen to represent significant differences.

# RESULTS

## Diatom resting membrane potential and membrane excitability are unaffected by PbTx-3

Under steady state conditions, *O. sinensis* cells exhibited endogenous membrane oscillations with spontaneous firing of rapid action potentials (50–100 ms duration) when the threshold voltage was reached ($n = 24$ cells, Figs. 1A and 1B). Long periods of very stable negative membrane potentials were also observed in some cells (Fig. 1C). As with animal $Na^+$ VGCs, the diatom action potentials comprised a fast rising phase followed by a slower hyperpolarization phase (Fig. S1). The cells either elicited single or several repetitive action potential spikes (Figs. S1A and Figs. S1B). In all cells examined, whether exhibiting

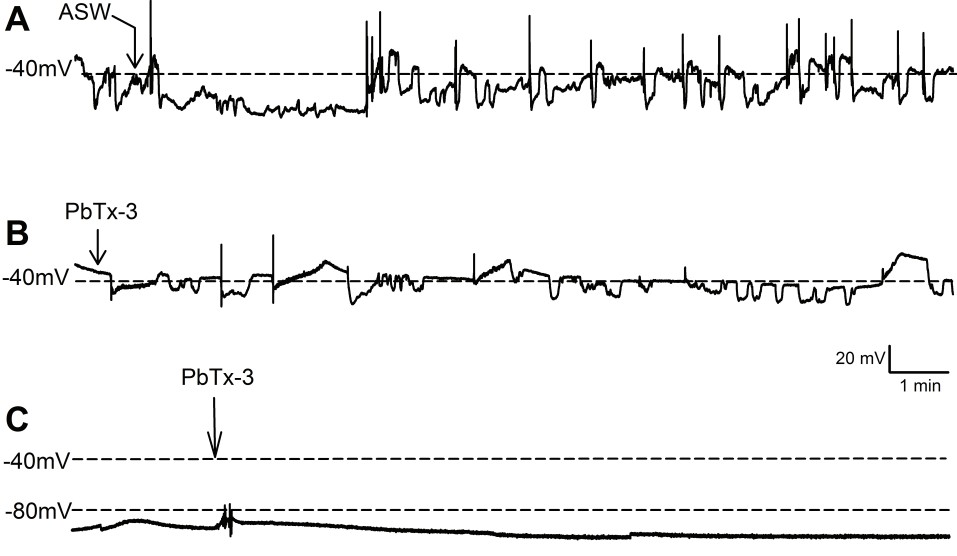

**Figure 1 Effect of PbTx-3 on *O. sinensis*. membrane potential.** Representative traces show free-running membrane potential monitored over 15 min following additions of (A) ASW and (B, C) 1 μM PbTx-3 indicated by arrows. No significant change in pattern of membrane potential was observed ($n = 4$ and 12 cells for control and treatments, respectively) for either cells exhibiting spontaneous oscillations (A, B) or quiescent membrane potential (C).

quiescent or oscillating membrane potentials, injections of 1 nA current over 10 ms evoked action potentials (Fig. S1C).

Bolus additions of ASW represented a control treatment and did not induce any deviation from the steady state recordings for up to 15 min ($n = 4$ cells, Fig. 1A). Additions of 1 μM PbTx-3 for 15 min also had no significant effect on free-running resting membrane potential in *O. sinensis* ($n = 12$ cells). This lack of response in the presence of PbTx-3 was consistent regardless of the pattern of membrane potential exhibited by any given cell, as cells exhibiting both spontaneous action potentials ($n = 7$ cells) and stable negative membrane voltages ($n = 5$ cells) are represented in the PbTx-3 treatments (Figs. 1B and 1C).

## PbTx-3 partially blocks *O. sinensis* Na⁺/Ca²⁺ currents

Voltage clamp experiments were conducted to investigate whether more subtle effects on VGCs were induced by interaction with PbTx. A depolarizing voltage-step protocol from a holding potential of $-100$ mV to 50 mV in 10 mV increments was used to activate $Na^+/Ca^{2+}$ currents (Fig. 2A). The control bolus additions of ASW ($n = 7$ cells) yielded no change in fast inward current or slower delayed outward current after a 15 min treatment (Fig. 2A). Addition of 1 μM PbTx-3 caused a $33.6 \pm 6.6\%$ decrease in average peak $Na^+/Ca^{2+}$ current amplitude from $-19.7 \pm 2.6$ nA to $-13.0 \pm 2.0$ nA ($n = 14$ cells, Student's $T$-test $p < 0.001$) after a 15 min treatment (Figs. 2A and 2B). Neither voltage activation nor voltage of the peak current was significantly affected by PbTx-3 over 15 min ($-47.2 \pm 3.8$ mV and $-45.7 \pm 3.0$ mV for control and PbTx-3 treatments, respectively; Fig. 2C). In addition to the suppressed peak inward current, PbTx-3 caused a distinct shift

in the reversal potential of the inward current from $15.0 \pm 2.6$ mV to $32.2 \pm 5.8$ mV ($n = 14$ cells, Student's $T$-test $p < 0.01$) suggesting a change in the selectivity of the underlying ion channels (Fig. 2C). Washout with toxin-free ASW perfusion on PbTx-3 treated cells showed limited recovery after 15 min (peak amplitude $-14.9 \pm 2.4$ nA, $n = 10$ cells, data not shown), and partial recovery after 30 min (peak amplitude $-17.6 \pm 6.1$, $n = 4$ cells, data not shown). Analysis of the delayed outward current showed a 25% decrease in current amplitude from $13.1 \pm 1.1$ nA to $9.98 \pm 0.76$ nA ($n = 14$ cells, Student's $T$-test $p < 0.001$, Fig. 2C).

Treatment with 1 mM lidocaine, known to block animal and diatom $Na^+$ currents, caused a $61.7 \pm 7.2$ % ($n = 4$ cells, data not shown) inhibition in the evoked $Na^+/Ca^{2+}$ VGC of *O. sinensis* which is consistent with previous data and confirmed that the perfusion protocol was sufficient to deliver toxin treatments during electrophysiological experiments (*Taylor, 2009*).

### Negligible effects of PbTx-3 on diatom VGC kinetics

PbTxs have been shown to modify both activation and inactivation properties of animal $Na^+$ channels (*Huang, Wu & Baden, 1984*; *Jeglitsch et al., 1998*). To investigate the effect of PbTx-3 on the activation kinetics of the *O. sinensis* $Na^+/Ca^{2+}$ VGCs, conductance was calculated according to *Strachan, Lewis & Nicholson (1999)*, normalized to peak conductance and fitted with a Boltzmann function in order to determine the half activation voltage ($V_h$) (Fig. 3A). No significant change in $V_h$ was observed between the control ($-58.5 \pm 1.0$ mV) and PbTx-3 treatments ($-57.9 \pm 1.8$ mV, Student's $T$-test $p > 0.05$).

The effects of PbTx-3 on the inactivation of *O. sinensis* $Na^+/Ca^{2+}$ current were determined using a voltage protocol of 50 ms pre-pulses from $-100$ mV to $-15$ mV in 5 mV increments applied before a depolarizing pulse selected to evoke a full inward current. The peak evoked current values were normalized to the maximum peak and plotted against pre-pulse voltage before fitting a Boltzmann function (Fig. 3B). A small but significant positive shift in the half-inactivation voltage ($V_{inact}$) was observed in the presence of PbTx-3 ($-65.7 \pm 0.5$ mV) compared to the control $V_{inact}$ ($-69.4 \pm 0.5$ mV, $n = 14$ cells, Student's $T$-test $p < 0.05$). A double pulse protocol was used to further characterize the possible effects of PbTx-3 on the inactivation properties of these channels (Fig. 3C). No significant change in recovery from inactivation was observed in PbTx-3 treatments (control $\tau = 5.1$ ms $\pm 0.4$ and PbTx-3 $\tau = 6.1$ ms $\pm 0.4$, Student's $T$-test $p > 0.05$, Fig. 3C).

### Brevetoxin binding site-5 analysis

Using available sequence resources from selected metazoans, alveolates (dinoflagellates, ciliates and *Perkinsus*) and stramenopiles (diatoms, brown algae, and oomycetes), putative VGCs were recovered from BLAST similarity searches (Table S1). Unexpectedly, most diatom sequences found using this approach, including *O. sinensis*, were single-domain VGCs with greatest similarity to the bacterial $Na^+$ VGC NaChBac (Table S1, Fig. 4A and Fig. S2) (*Ren et al., 2001*). Only one diatom, *T. pseudonana*, appeared to possess both single-domain and 4-domain $Na^+/Ca^{2+}$ VGCs (Figs. 4A and 4B).

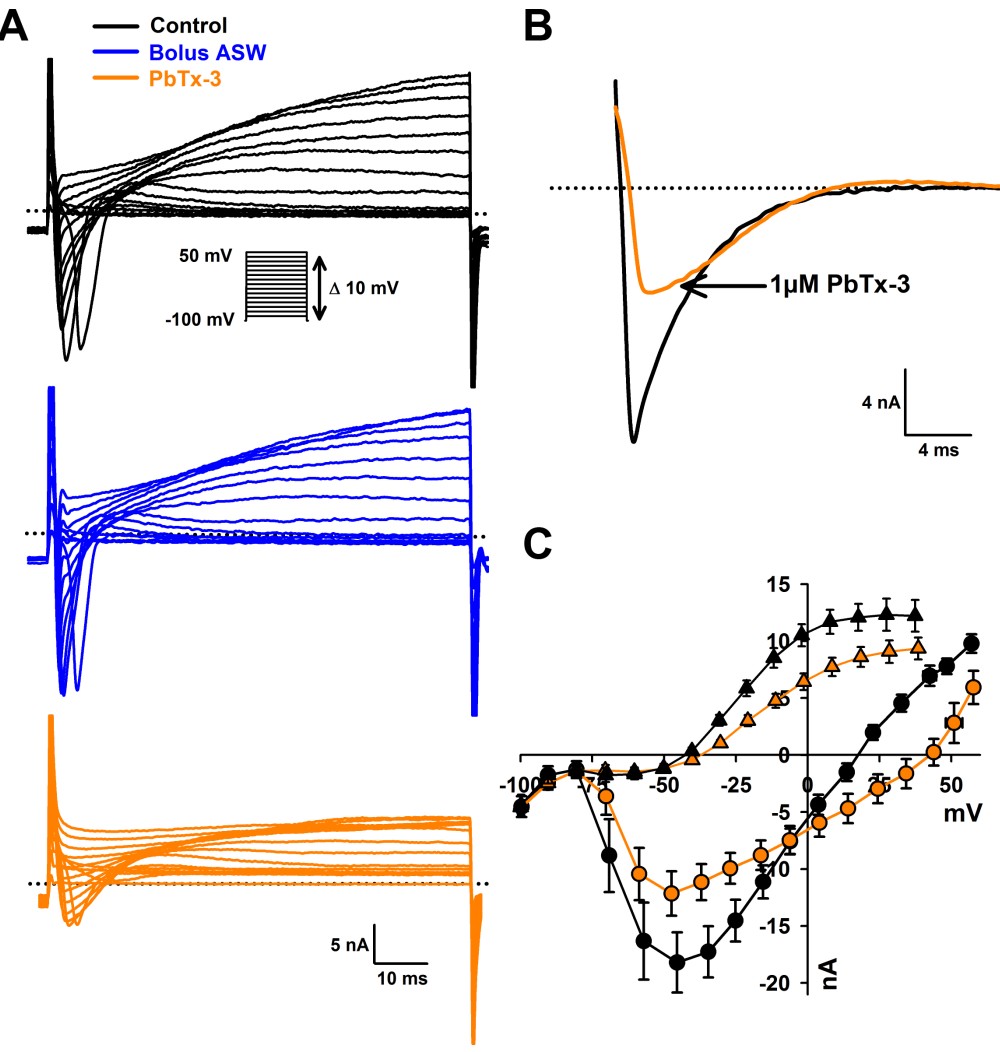

**Figure 2  Effects of PbTx-3 on *O. sinensis*. voltage activated currents.** (A) Representative ion currents evoked in response to depolarizing voltage clamp pulses under control conditions, after 15 min treatment of bolus ASW ($n = 7$ cells tested) and after treatment with 1 μM PbTx-3 for 15 min showing a clear decline in both peak inward and delayed outward currents ($n = 14$ cells tested). (B) Detail of fast inactivating inward current evoked by voltage clamp pulse from −100 mV to −60 mV before and after 1 μM PbTx-3 perfusion. (C) Combined current-voltage plot for PbTx perfusion experiments in which average peak inward current was plotted as a function of membrane voltage in the absence (black circles) and presence of 1 μM PbTx-3 (orange circles), and peak outward current in the absence (black triangles) and presence of 1 μM PbTx-3 (orange triangles). $n = 14$ experiments. Standard error bars are indicated.

To investigate whether a previously characterized PbTx-3 binding site could be found in the putative $Na^+/Ca^{2+}$ VGCs of SAR-group (Stramenopiles, Alveolates and Rhizaria), a multiple sequence alignment was used (Fig. 4C). The characterized site-5 in metazoan $Na^+$ VGCs comprises two 18–20 amino acid segments, the first directly following the conserved S6 of domain 1 (D1), and the second, follows S5 of domain 4 (D4) (Fig. 4A) (*Trainer, Baden & Catterall, 1994*). Among all of the representative eukaryotic 4-domain VGC sequences and concatenated sequence of single-domain VGC from *O. sinensis* (#

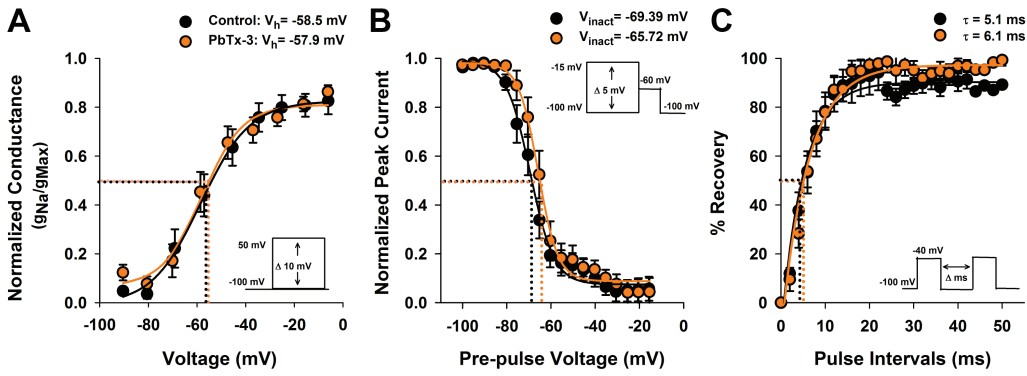

**Figure 3** **Activation and inactivation properties of diatom Na⁺/Ca²⁺ currents in the presence of PbTx-3.** (A) Voltage activation was determined by calculating the peak Na⁺/Ca²⁺ conductance (g) for each voltage pulse and normalized to the maximum conductance. The resulting g-V values were fitted to a Boltzmann distribution. Half activation voltage ($V_h$, the voltage stimulus that evoked 50% of the peak current) was not significantly affected by 1 µM PbTx-3 (orange circles). (B) Steady state voltage inactivation was determined using a series of depolarizing voltage pre-pulses from a holding voltage of −100 mV were applied before evoking maximal inward currents by a test pulse to −60 mV. The average peak currents in response to test pulses were normalized to the maximum peak current and fitted to a Boltzmann curve. Half inactivation voltage ($V_{inact}$) was the pre-pulse voltage at which half the peak current was suppressed. There was a small but significant ($p < 0.05$, Students $T$-test) positive shift in the inactivation in the presence of PbTx-3 (orange circles) compared to controls (black circles). (C) Recovery from inactivation was measured using a double pulse protocol under control conditions in which the first voltage evoked a maximal inward Na⁺/Ca²⁺ current followed by a variable time interval (Δ ms), after which a second depolarization pulse was applied. The peak response to the second voltage clamp pulse after a 2 ms delay was less than 10% of the peak, demonstrating the majority of VCGs were inactivated. Delay of 10–20 ms was generally required for the current to be fully recovered. The same recovery from inactivation protocol was applied to the same cell after 1 µM PbTx-3 treatment showed a similar pattern of recovery although overall a decreased peak current amplitude as demonstrated in Fig. 2. Capacity transients are removed from the current traces for clarity. A recovery from inactivation plot where the second peak current response is expressed as a % of the first and plotted against the pulse time interval. Control (black circles) and 1 µM PbTx-3 (orange circles) curves were fit with an exponential with no significant difference in the time constant. In all experimental trials ($n = 14$) the standard error bars are indicated.

4108) into a 4-domain hypothetical homotetramer, D1–S6 and D4–S5 are moderately conserved across all taxa (Fig. 4C). However, while vertebrates showed strong conservation of site-5 compared to the mammalian Na$_v$1.2 channel, divergence of site-5 was apparent in invertebrate 4-domain VGCs of the squid *H. bleekeri* and the sea anemone *Exaiptasia pallida*, as well as in the choanoflagellate *Monosiga brevicollis*. The representative taxa from the SAR-group showed little sequence conservation at this characterized site-5.

## DISCUSSION

In squid, crayfish, frog and mammalian neuronal models it is known that 100 nM to 1 µM PbTx interacts with 4 domain Na⁺ VGCs causing hyperexcitability resulting in rapid firing of action potentials (*Kim et al., 1975*; *Westerfield et al., 1977*; *Huang, Wu & Baden, 1984*). The underlying cause is a −10 mV shift in voltage-dependent activation of Na⁺ VGCs (*Atchison et al., 1986*; *Sheridan & Adler, 1989*) and delayed inactivation (*Huang, Wu & Baden, 1984*; *Baden, Rein & Gawley, 1998*; *Jeglitsch et al., 1998*). In the present study,

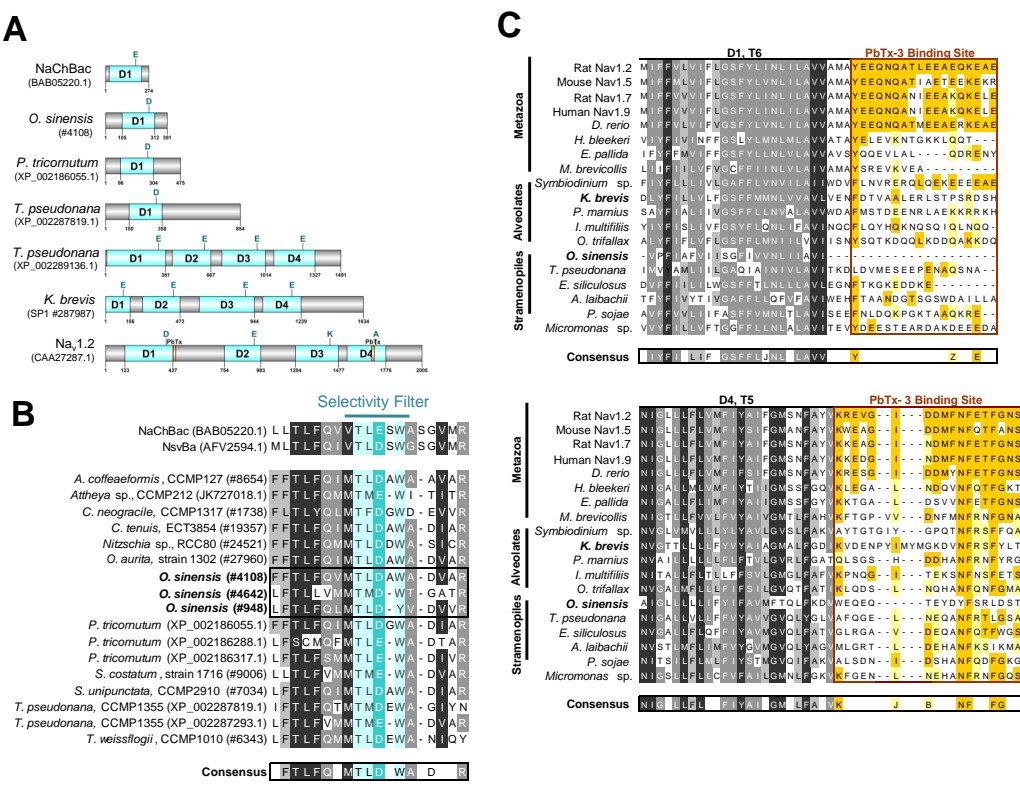

**Figure 4 Sequence analysis of diatom Na⁺/Ca²⁺ VGCs and PbTx binding site.** (A) Representative single domain Na⁺/Ca²⁺ VGCs were selected from bacteria (NaChBac), and the diatom *O. sinensis*, *P. tricornutum* and *T. pseudonana*. Four-domain channels are represented by the diatom *T. pseudonana*, dinoflagellate *K. brevis*, and mammalian Na$_v$1.2. The major trans-membrane spanning domains colored in blue. The site of the high field strength residue within the selectivity filter is indicated by a letter above each domain. In the 4-domain VGCs, the inactivation gate is indicated and the PbTx binding site previously identified in Na$_v$1.2 are highlighted in orange. (B) A multiple sequence alignment of the diatom single-domain channels obtained by BLAST searches with the bacterial channel NaChBac. The selectivity filter is indicated in light blue and the high field strength residue in teal. (C) Multiple sequence alignment of eukaryote 4-domain Na⁺ VGCs highlighting the two PbTx-3 binding regions characterized by *Trainer, Baden & Catterall (1994)*. Conserved residues are labeled in orange (80–100% similarity) and yellow (80–60% similarity). Highly conserved residues of the upstream transmembrane segments are colored based on similarity using the Blosum62 score matrix (*Henikoff & Henikoff, 1992*), where darker shades of grey are more similar. Sequences were obtained from NCBI and iMicrobe databases with further details provided in Table S1.

*O. sinensis* Na⁺/Ca²⁺ VGCs did not exhibit these characteristic changes in function when exposed to 1 μM PbTx-3, but did exhibit an interaction that resulted in reduced permeation and a positive shift in reversal potential. The reduction in the peak inward Na⁺/Ca²⁺ current in *O. sinensis* was similar to that observed in squid axons (*Huang, Wu & Baden, 1984*), but was not accompanied by significant changes in kinetics of the current or membrane hyperexcitability. It has been suggested that PbTxs may alter subconductance states reflecting a protein conformation in which the flow of ions through the channel pore is less efficient (*Jeglitsch et al., 1998*). The slower outward rectifying current of *O. sinensis* associated with the repolarizing phase of the action potential was also inhibited by the

PbTx-3, which indicates that at these relatively high concentrations, inhibitory interactions of PbTx may not be channel specific.

The PbTx-3-induced positive shift in the reversal potential of $Na^+/Ca^{2+}$ currents in *O. sinensis* implies a change in ion selectivity in favor of $Ca^{2+}$ ions. The *O. sinensis* VGCs that underlie the diatom action potential are known to be permeable to both $Na^+$ and $Ca^{2+}$ (*Taylor, 2009*), presumably coupling membrane excitability with intracellular signaling processes through changes in cytoplasmic $Ca^{2+}$ (*Falciatore et al., 2000*; *Vardi et al., 2006*). PbTx-3 interaction at the pore of the channel may directly or allosterically alter channel selectivity, which would result in altered $Ca^{2+}$ influx and associated intracellular $Ca^{2+}$ dynamics (*LePage, Baden & Murray, 2003*). However, the permeability shift of the whole cell currents could also be attributed to PbTx inhibition of a subpopulation of $Na^+/Ca^{2+}$ channels with lower $Ca^{2+}$ permeability that would otherwise contribute to the whole channel current. PbTx inhibition and altered reversal of $Na^+/Ca^{2+}$ whole cell currents has not to our knowledge been previously documented for animal $Na^+$ selective currents, and appears to be a unique feature of PbTx interaction with the diatom $Na^+$ VGC. Given the key role that intracellular $[Ca^{2+}]_{cyt}$ plays in mediating stress responses of phytoplankton, including programmed cell death (see *Bidle (2015)* and references therein) it will be interesting to determine whether PbTx interactions with phytoplankton $Na^+/Ca^{2+}$ channels affects intracellular $Ca^{2+}$ dynamics at steady state or in response to biotic or abiotic stressors.

The lack of effect of PbTx-3 on the kinetics of activation and inactivation of the *O. sinensis* $Na^+/Ca^{2+}$ currents points to a novel interaction of these molecules with the underlying algal VGCs. This is consistent with the absence of hyperexcitability observed in the presence of PbTx-3 that would be expected if voltage dependent activation and inactivation were shifted negative. Overall, PbTx-3 effects were observed on permeation and peak amplitude of diatom currents, but were not accompanied by significant changes to their kinetic properties. We conclude that *O. sinensis* VGCs may be affected by PbTx-3 interactions, albeit with reduced efficacy compared to PbTx-sensitive metazoan channels (*Westerfield et al., 1977*; *Huang, Wu & Baden, 1984*; *Jeglitsch et al., 1998*). Under normal bloom conditions levels of PbTxs in surface waters are rarely observed greater than 30 to 80 nM (*Backer et al., 2005*; *Tester et al., 2008*), and although longer term effects or species-specific responses (*Kubanek et al., 2005*; *Poulson et al., 2010*; *Poulson-Ellestad et al., 2014a*) cannot be ruled out, it seems unlikely, at least in the case of *O. sinensis*, that VGCs and interference of diatom action potential mediated processes are significantly impacted by waterborne exposure to toxins produced during blooms of *K. brevis*. This is consistent with culture-based experiments on a sister species of *O. sinensis* from the Gulf of Mexico, *Odontella aurita*, that did not experience reduced cell growth in the presence of live *K. brevis* but instead significantly inhibited growth of *K. brevis* (*Kubanek et al., 2005*; *Poulson-Ellestad et al., 2014b*).

The potential for PbTx-3 to interact with diatom $Na^+/Ca^{2+}$ VGCs was investigated by sequence comparison of the PbTx binding site of 4-domain $Na^+/Ca^{2+}$ VGCs among a range of taxa including representatives from the SAR-group. There is no clear conservation of amino acid residues across site-5 in this group, although there are more conserved

residues in the extracellular portion of site-5. In animal Na$^+$ VGCs, that are PbTx-sensitive, biophysical evidence shows the toxin interacts with the pore domain at both the intracellular and extracellular side, thereby affecting permeation and inactivation kinetics (*Huang, Wu & Baden, 1984*). The lack of conservation of site-5 in the SAR-group 4-domain VGCs implies PbTx interactions are likely compromised which could explain their relative insensitivity to this toxin. Nevertheless, given the flexibility of the PbTx molecule, it is not unreasonable to speculate that novel biophysical outcomes of PbTx may occur as a result of binding to the more conserved extracellular receptor site, and interaction with alternate sites within the α-subunit of the channel protein altering Na$^+$ VGC function (*Trainer, Baden & Catterall, 1994*). Interestingly, our analysis of site-5 also showed poor homology in the D1T6 part of site-5 of the squid *H. bleekeri* Na$^+$VGC homolog although previous biophysical studies have used *H. bleekeri* axons as a model system to demonstrate the highly specific biophysical effects of PbTx on Na$^+$ VGC function (*Kim et al., 1975*; *Westerfield et al., 1977*; *Huang, Wu & Baden, 1984*). Therefore, the D4T5 region of site-5 may be the most important residues for PbTx interaction. It is also possible that the amino acid residues in rat brain Na$_v$1.2 from which site-5 was determined may not represent the only PbTx interaction site, especially in VGCs of lower invertebrates and other eukaryotes.

An alternative explanation for relative insensitivity of *O. sinensis* Na$^+$/Ca$^{2+}$ currents to PbTx is that this species appears to lack a 4-domain VGC (Fig. 4), as has also been reported for the genome of the pennate diatom *Phaeodactylum tricornutum* (*Verret et al., 2010*). An unexpected finding of the present study was the prevalence of expressed single-domain Na$^+$/Ca$^{2+}$ VGCs (similar to single domain bacterial channels, NaChBac) in diatom transcriptomes. Only one putative 4-domain Na$^+$/Ca$^{2+}$ VGC was found in the genome of centric diatom *T. pseudonana* that was previously described (*Taylor, 2009*; *Moran et al., 2015*). This is in marked contrast to other algal lineages including chlorophytes such as *Chlamydomonas* (*Fujiu et al., 2009*) and *Micromonas* (*Verret et al., 2010*), the multicellular stramenopile *Ectocarpus* (*Verret et al., 2010*), and in *K. brevis* itself that reportedly has seven 4-domain VGCs predicted from its transcriptome (*Ryan, Pepper & Campbell, 2014*) that each bear the Ca$^{2+}$ selective permeability motif (EEEE, Fig. 4A.) suggesting an important role in Ca$^{2+}$ transport and signaling.

Single-domain bacterial Na$^+$ VGCs comprise six transmembrane helices (S1-6) with a voltage sensor of arginine residues within S4 and pore loop between S5 and S6. These bacterial VGCs have been shown to be involved in motility, chemotaxis, pH homeostasis and metabolic growth (*Ito et al., 2004*; *DeCaen et al., 2014*). They form homotetramers and when expressed in heterologous systems show many of the biophysical features of eukaryote 4-domain VGCs (*Ren et al., 2001*; *Koishi et al., 2004*; *Scheuer, 2014*). The large 4-domain VGCs of eukaryotes form two major groups, the Ca$^{2+}$ and Na$^+$ VGCs, that were proposed to have arisen from gene duplication and undergone functional diversification (*Strong, Chandy & Gutman, 1993*; *Cestèle & Catterall, 2000*). Shared structural features between eukaryotic and prokaryotic VGCs lead to the hypothesis that VGCs ultimately originated from the ancestral single-domain prokaryote channel (*Anderson, Roberts-Misterly & Greenberg, 2005*), although, recent phylogenetic studies indicate independent

origins (*Verret et al., 2010*; *Liebeskind, Hillis & Zakon, 2013*). However, in both bacterial and eukaryote VGCs the high field strength (HFS) residue in the pore loop between S5 and S6 determines ion selectivity and more recent structural and functional analyses suggests additional sites downstream in the ascending pore loop may also play critical roles in ion permeation (Fig. S2) (*Ren et al., 2001*; *Liebeskind, Hillis & Zakon, 2013*; *DeCaen et al., 2014*; *Stephens et al., 2015*). In vertebrates the $Na^+$ selectivity filter HFS sites surrounding the pore domain creates a heterotetrameric DEKA motif (Fig. 4A), but in bacteria the motif is composed of glutamate residues from each of the four monomers similar to the $Ca^{2+}$ EEEE filter (*Finol-Urdaneta et al., 2014*). Even though the EEEE selectivity motif is more similar to eukaryotic 4-domain $Ca^{2+}$ VGCs, bacterial NaChBac are strongly selective for $Na^+$ over $K^+$ or $Ca^{2+}$ (*Yue et al., 2002*; *Koishi et al., 2004*; *Scheuer, 2014*). Recent studies show that point-mutation of the bacterial $Na^+$ selectivity filter (TL*E*SW → TL*D*SW) can induce non-selective cation transport (*DeCaen et al., 2014*; *Finol-Urdaneta et al., 2014*; *Tang et al., 2014*). The single domain VGCs found in diatoms possess a mixture of $Na^+$ selective and non-selective filter motifs (Fig. 4B), which could explain their permeability to both $Na^+$ and $Ca^{2+}$ ions (*Taylor, 2009*). The widespread presence of single domain VGCs in diatoms raises a number of interesting questions including; whether they functionally substitute for 4-domain channels in generating diatom action potentials, and what selective pressures led to apparent loss of 4-domain $Na^+/Ca^{2+}$ VGCs in diatoms.

## CONCLUSIONS

In summary, *O. sinensis* membrane potential did not demonstrate the characteristic hyperexcitabilty associated with PbTx exposure. However, evoked voltage activated currents that underlie the diatom action potential were partially inhibited and accompanied by an apparent shift in the ion selectivity that favors $Ca^{2+}$ influx. Nevertheless, the apparent lack of 4-domain eukaryotic VGCs in *O. sinensis* and poor conservation of the PbTx binding site could underlie this unique interaction. Given the relatively weak effect of PbTx, *O. sinensis* VGCs and cell signaling are likely to be relatively insensitive to PbTx at environmentally relevant concentrations. Further examination of sympatric phytoplankton species, specifically those with documented allelopathic interactions (*Kubanek et al., 2005*; *Poulson et al., 2010*) is necessary to elucidate PbTx interactions with diverse algal VGCs and their signaling functions.

## ACKNOWLEDGEMENTS

We would like to thank J Craig Bailey and Carmelo Tomas for their helpful discussions and editorial comments during manuscript preparation. We would also like to thank Elizabeth Elliot for technical assistance on the project.

### Funding

This work was supported by a University of North Carolina Wilmington Center for Marine Science Pilot Grant and by the National Science Foundation (IOS 0949744). The funders had no role in study design, data collection and analysis, decision to publish, or preparation of the manuscript.

### Grant Disclosures

The following grant information was disclosed by the authors:
University of North Carolina Wilmington Center for Marine Science Pilot Grant.
National Science Foundation: IOS 0949744.

### Competing Interests

The authors declare there are no competing interests.

### Author Contributions

- Sheila A. Kitchen and Alison R. Taylor conceived and designed the experiments, performed the experiments, analyzed the data, prepared figures and/or tables, authored or reviewed drafts of the paper, approved the final draft.
- Andrea J. Bourdelais contributed reagents/materials/analysis tools, authored or reviewed drafts of the paper, approved the final draft.

### Data Availability

The raw data are provided as Supplemental Files.

### Supplemental Information

Supplemental information for this article can be found online at http://dx.doi.org/10.7717/peerj.4533#supplemental-information.

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
