# Peer review of "Interaction of a dinoflagellate neurotoxin with voltage-activated ion channels in a marine diatom"

_PeerJ, doi:10.7717/peerj.4533_

## Round 0.1 · original submission · Minor Revisions

--- start smd text ----

Dear Dr. Taylor

your paper has been reviewed by 3 experts in the field. I am very impressed by the effort they have put into providing constructive comments on your obviously excellent work. Your paper is returned to you with request for minor revisions, although some of the comments are not minor. I ask you to very seriously consider the feedback and incorporate it in a revised manuscript, which will very likely be acceptable. Please also provide a detailed response to the reviewers comments, especially in cases where you disagree with their assessment.

Best
smd

PS. I am writing this text, which will be embedded within a pre-formatted email from PeerJ. I can not edit this. So I apologize if the email as a whole seems confusing.

--- end smd text ----

Reviewer 1 ·

Basic reporting

The basic reporting of this paper was expertly done. The writing was clear and unambiguous. The raw data was shared in the supplemental materials section, literature cited was current and provided good context. The structure of the article was great, with professional looking figures.

Experimental design

On the whole, the manuscript does a good job describing the methods and presenting their research questions. That said, I have minor issues with some parts of the methodology.


1. Was the diatom culture axenic?

2. Line 143 - it is unclear what is meant by "exponential growth phase is determined by cell counts (10^5) with a..." does this mean that the authors counted 10^5 cells? That was the final concentration? This needs to be clarified.

3. On Line 175 - "The perfusion was then stopped and a bolus addition of ASW...." what volume were these bolus additions added in?

Validity of the findings

The findings on the whole were presented well and the conclusions reached were well thought out. Minor issues:

1. Just a comment that it would have been nice to show the impact of the PbTx3 toxin on long term growth of the diatom culture. While you don't see huge differences in terms of hyperexcitability, there are differences in the action potential. It would have been nice to see if that indeed translated into differences in growth rates of the cells.

2. Interesting finding that the action potential inhibition was accompanied by an apparent shift in the ion selectivity that favors Ca2+ influx. The influx of Ca2+ into a cell is a typical stress response that has been observed in phytoplankton previously (see refs in Bidle 2015). Without additional physiological information (staining for intracellular Ca2+ or just diatom growth) it is hard to know what the long-term impacts are, but Ca2+ signaling has been suggested to play a role in programmed cell death, so it could be that this is a significant response of the cell, but with no information for how the cell survives exposure it is hard to draw conclusions. Inclusion of some discussion of Ca2+ signaling in phytoplankton could be added to the discussion.

3. The final statement on Line 329 - 331, seems a bit heavy handed. In reality one toxin was tested in these experiments, while the authors state that it is the toxin most prevalent, certainly diatoms could react to the other forms of the PbTx toxin or other waterbourne chemical cues.

Additional comments

Nice paper, it is nice to see people trying to look for the mechanisms of action in response to toxin or secondary metabolite exposure.

Reviewer 2 ·

Basic reporting

Well-written manuscript with clear, unambiguous and professional English. Literature review is adequate, but see general comments with respect to breadth of discussion.

Note comments on raw data format in general comments.

Study is self-contained, but linkages between electrophysiology and bioinformatics could be strengthened.

Experimental design

Research question is well defined and falls within aims and scope of the journal. Knowledge gap clearly identified in introduction.

There is a mismatch between the motivating hypothesis and the study details (see general comments). Additional methodological details needed for replication of results.

Validity of the findings

The findings seem to be valid, but more experimental details are needed to properly assess bioinformatic findings (see general comments).

Additional comments

Overview

In the introduction, the authors motivate their study with the hypothesis that toxin production by species like Karenia brevis may function as allelopathic compounds that inhibit growth in potential competitors. This is a compelling hypothesis. While much work has been completed in elucidating the effects of neurotoxins on ion channel function, little progress has been made in establishing the competitive advantages that the toxins confer to the algal species producing them.

The experimental approaches in this study were two-fold: 1) electrophysiological measurements on the effect of brevetoxin on channel function; and 2) bioinformatic analysis comparing the amino-acid structure of brevetoxin binding sites of voltage-gated channels from diatoms, other phytoplankton species, bacteria and “animals”. While I found the study very interesting, I did not think it answered the question set up in the introduction. Specifically, in the experimental treatment, a single (and high; 1 µm) concentration of brevetoxin was applied to the cells. A better experimental design would have been to use a range of concentrations, including concentrations that would have been closer to environmental levels (30 - 80 nm; line 327). It seems that a simpler approach would have been to grow the cells in the presence/absence of brevetoxin to establish allelopathy.

Nevertheless, I found the study very interesting, in particular the combination of physiological measurements with a bioinformatic approach. The effect of the toxin on the channel function is novel. The similarity/dissimilarity of VGCs across different taxa has been an active area of research, including the relatively recent conclusion of independent evolution of the bacterial and the eukaryotic cation channels (e.g., McCusker et al. 2012). Thus, the presence of bacterial channels in the diatoms is interesting, although the current manuscript does not indicate whether they are expressed (see comments on bioinformatics).

I had some concerns about the electrophysiological methods. Given the unusual structure of the diatom, the presence of large vacuoles and the likelihood of recordings from multiple channels, the materials and methods need to include more details to allow a better assessment on what is being recorded, data filtering and analysis.

Discussion – much of the discussion is too speculative and needs to be revised. Specifically, the comparisons with the physiology of vertebrate and invertebrate Na+/Ca++ channels needs to be much more specific – throughout the discussion there are overgeneralizations mentioning “animal” channels, which may be only applicable to certain species/channels and an oversimplification of the these channels, and thus not represent the diversity of VGCs in vertebrates and invertebrates.

The TTX story may provide an interesting framework for interpretation of the current results. Using site-directed mutagenesis, it has been possible to gain insights into how conformational changes caused by single amino acid mutations affect the TTX binding site. While brevetoxin may not have been studied as extensively, brevetoxin resistance has been characterized for some channels, and references to this literature are incomplete (e.g., Dechraoui MY, Wacksman JJ, Ramsdell JS., 2008).

Bioinformatic analysis

Provide more details on the searches performed including the specific databases in NCBI that were searched. Depending on the database, it is quite possible that only partial sequences were available for some species. Include a description of the criteria used (such as reciprocal blasts) to vet the sequences that were retrieved using the blast searches. Search results should include length of the sequences retrieved, whether proteins are full length, top hits, e-values.

No information is given whether the searches were limited to genomic data with predicted protein sequences, or if they also included mRNA (transcribed) sequences or even mass spec analyses of the actual proteins (model species). Thus, the table should include information on the source of the predicted proteins, and the text provide an assessment of the databases searched and sequencing technology used, since the proteins are often predicted from incomplete information. Furthermore, the NaV1 sequence is particularly difficult to assemble from short sequence reads or to obtain using a PCR-based approach, due to regions of low complexity and splice variants. It is difficult to evaluate the quality of the bioinformatics results without additional information.

One issue is whether the absence of a NaV1 gene is due to incomplete sequence data or the true absence of the gene in the organism. This can only be addressed in species with genomes that have been completely sequenced, assembled and supported by extensive transcriptomic data to confirm transcription. The complete genomes of two diatoms have been published (Phaeodactylum tricornutum and Thalassiosira pseudonana) – these genomes could be searched to determine which genes would be expected in the databases. Thus, Fig. 4A should include the VGCs found in P. tricornutum as well. Are there more than 1VGC gene present in these diatoms? Table 1S would be more informative if it included information on e-values and top hits (bacterial vs. NaV1 vs. NaV2). Given the general audience of PeerJ, provide information on the phylogenetic relatedness between the study organism and these two diatoms.

Raw data check

The data are included with the manuscript. However, the electrophysiological recordings are in “.abf” format (Axon Instruments?), which I could not read. Could these files be converted to “.cvs” format (or another format that is broadly accessible)? This would make them more broadly available.

Table S1 caption – provide a more detailed description of the data (see comments on bioinformatics section).

Table S1 – For many sequences the authors list the publication associated with the sequence, which I think is important. However, it may be useful to add the source of the sequences (which is only given in the absence of a publication). This would allow the reader to go directly to the appropriate website (e.g., NCBI) and retrieve the sequence. However, while the NCBI sequences were easy to find, I could not locate the sequences on “imicrobe”. Specifically, my searches indicated that the “imicrobe.us” website could not be found.

Figure S2 – needs to be checked for errors. At least two species are identified as CCMP127. Furthermore, it would be helpful to make it easier to cross-reference between this figure and Table S1 and Fig. 4, consider using a numbering system that is common to table and figure. This numbering system could be used to cross-reference with the sequences aligned in Figure 4B, C and D, which is not possible in the current format.

Specific comments
Line 221 – remove “in”

Line 291 – “animal models” – be more specific here, which animal models? Throughout the discussion avoid generalizing to “animal” – the diversity of ion channels is very large, and there are many taxonomic differences in gene duplication and function. Comparisons should specify the model species and the specific genes. NaV channels are characterized by the DEKA motif (which lends them their specificity), which is not true for the diatom channel, raising the question whether it should be considered a true Na+ channel – it seems like it is a Ca++ channel, that is also permeable to Na+ ions.

Lines 297-300 – the authors indicate that the response observed in their diatom is unique, yet seems to be similar to the squid (which is a standard animal model). This section is unclear: how is the diatom’s response similar/unique, and how does the squid compare with other “animal models”.

Paragraph starting with line 332 – is too speculative. Needs to be rewritten after revision of the bioinformatics section. It seems to be premature to speculate on the evolution of the channels given the diversity in amino acid sequences shown in Figure 4. A lot is known about the evolution of VGC channels in invertebrates and vertebrates, and channel function – but this is not reflected in this paragraph. Furthermore, the wealth of information available on vertebrates and invertebrates stands in contrast to the limited understanding of VGC function in bacteria and unicellular organisms like the diatoms.

Paragraph starting with line 354 – the conclusion of the lack of a 4 domain VGC is not supported by the current result section (see figure 4C and D with sequences shown for O. sinensis). Note comments on bioinformatics. This paragraph also seems to be contradictive – if the EEEE filter is Ca++ specific (bacterial channel), and the DEEE filter is a general cation channel, what is the basis for identifying the predicted diatom DDDD motif as being a selective Na+ channel?

Reviewer 3 ·

Basic reporting

Kitchen et al studied the impact of the Karenia breve neurotoxin (Brevetoxin) on marine phytoplankton as they hypothesize that these neurotoxins may also affects the sodium and calcium VGCs in marine phytoplankton like in animals. This is an obvious hypothesis to test, but not much was found in the literature that brevetoxins acting as allelopathic subtances, see Cembella 2003 and others.
However, the MS is clear and well written and the experimental set-up follow a logical structure and is well conducted. The combination of electrophysiology to analyse the VGC performance and the molecular genetic analyses of the potential VGC structure and function in phytoplankton such as the diatom Odontella presents a new level of effect analyses.
I wonder if the MS would benefit when the authors would put the insensitivity of diatoms and other phytoplankton species towards PbTx-3 into the focus and uses the results presented here to discuss a potential evolutionary step by changing the 4-domain channel losing the PbTx binding side and may using a single domain channel.. As the literature, so far as I know, was suggesting rather insensitivity of phytoplankton towards PbTx I was not expecting effects, but the results presented here serving with a nice dataset explaining why they are not sensitive. Different structure of the 4 domain channel and additional the one domain type. Anyway, just my thoughts and if I got it right, I think it should be explained more schematic so it will be easier to follow.
Introduction and background: The first paragraph of the introduction presents a good motivation for the model but I am wondering if not more recent publications can be used. For instance the increased frequency and duration has valid references from ~2000 and I think that there are more recent ones, particular from the group of Van Dolah and others.
Line 70-77: There is a new van Dolah paper out (J. Phycol. 53, 1325–1339 (2017)) about some new findings according the biosynthetic pathways and related PKS genes.
L80: “…it takes only nM concentrations of these potent toxins”…..Please be aware that this might be not much for an animal accumulating toxins over time, but in the water column, it is a lot. I have no information that any phytotoxin was measured in such high concentration in water. For the allelopathy hypothesis the effect should be in pM range I guess. The part could be rewritten.
L82-86: These sentences are a bit misleading as they suggesting/indicating that brevetoxins (…”these algal secondary metabolites..”) show allelopathic effects. Maybe it would be better to restructure the whole paragraph and put this general idea as the starting point and ask if PbTx can have similar functions…..with the following part (L91-96) you may support the hypothesis.
L91-96: These sentences are the most important ones to support the hypothesis of the MS. I would appreciated to get here more details of the organisms and effects.

Results: L287/L334: The chromalveolates were not a natural group and therefore the RAS-group (Rhizaria, Alveolates, Stramelopiles) are a better descriptor.
Figures are okay. I may got confused while reading the MS about the model to explain the insensitivity. I do not know how, but maybe the Figure 4 can be used to point out how the sequence differences in the four domain channel and the one domain channel support the conclusion.

Experimental design

So far I can comment on it, the methods are described with sufficient detail and informations. I cannot comment on the electrophysiology.

Validity of the findings

I appreciate the conclusion and the model which may help to explain why some phytoplankton species are insensitive to PbTx. As mention above, in my opinion that should be stronger highlighted.

---

## Round 0.2 · accepted · Accept

Dear Dr. Kitchen

Thank you for your thoughtful and thorough replies to the reviewers feedback and implementation in your manuscript.

Best
smd